# Pre-critical fluctuations and what they disclose about heterogeneous crystal nucleation

Martin Fitzner [1], Gabriele C. Sosso[2], Fabio Pietrucci[3], Silvio Pipolo[4] & Angelos Michaelides[1]

Heterogeneous crystal nucleation is ubiquitous in nature and at the heart of many industrial applications. At the molecular scale, however, major gaps in understanding this phenomenon persist. Here we investigate through molecular dynamics simulations how the formation of precritical crystalline clusters is connected to the kinetics of nucleation. Considering heterogeneous water freezing as a prototypical scenario of practical relevance, we find that precritical fluctuations connote which crystalline polymorph will form. The emergence of metastable phases can thus be promoted by templating crystal faces characteristic of specific polymorphs. As a consequence, heterogeneous classical nucleation theory cannot describe our simulation results, because the different substrates lead to the formation of different ice polytypes. We discuss how the issue of polymorphism needs to be incorporated into analysis and comparison of heterogeneous and homogeneous nucleation. Our results will help to interpret and analyze the growing number of experiments and simulations dealing with crystal polymorph selection.

[1] Thomas Young Centre, London Centre for Nanotechnology and Department of Physics and Astronomy, University College London, Gower Street London, London WC1E 6BT UK. [2] Department of Chemistry and Centre for Scientific Computing, University of Warwick, Gibbet Hill Road, Coventry CV4 7AL UK. [3] Institut de Minéralogie, de Physique des Matériaux et de Cosmochimie, CNRS UMR 7590, IRD UMR 206, MNHN, Sorbonne Universités—Université Pierre et Marie Curie Paris 6, F-75005 Paris, France. [4] Université de Lille, CNRS, Centrale Lille, ENSCL, Université d' Artois UMR 8181— UCCS Unité de Catalyse et Chimie du Solide, F-59000 Lille, France. Correspondence and requests for materials should be addressed to A.M. (email: angelos.michaelides@ucl.ac.uk)

Freezing of a liquid is typically initiated through contact with a foreign material as the homogeneous (hom.) barrier for crystal nucleation can be exceedingly large[1]. Therefore, heterogeneous (het.) nucleation is at the heart of a variety of processes like intracellular freezing[2] or the formation of amyloid fibrils in the brain that are related to diseases like Alzheimer's[3]. The theoretical understanding of het. nucleation is thus important to many branches of science and technology, ranging from pharmaceuticals[4,5] to cloud physics[6,7] to crystal engineering aimed at realizing novel materials discovered with theoretical approaches such as the materials genome initiative[8]. Although the first successful explanations of het. nucleation date back almost a century[9,10], there are many aspects of this phenomenon that still remain elusive[11,12]. To complement the growing experimental effort[13–16], computer simulations have proven helpful in uncovering fundamental aspects of the nucleation process[17–23], and are becoming increasingly useful in screening different substrate types and shapes to rank and understand their ability to enhance nucleation[24–28].

Despite its flaws[12,29], classical nucleation theory[30,31] (CNT) provides a qualitative understanding of nucleation, and due to its simplicity is still the most widely used theoretical framework to interpret experiments and simulations. The free energy cost for a crystalline cluster of size $n$ in heterogeneous classical nucleation theory (hetCNT) is given by:[1]

$$F(n) = f_V \cdot \Delta F \left[ 3 \left( \frac{n}{f_V \cdot n_c} \right)^{2/3} - 2 \left( \frac{n}{f_V \cdot n_c} \right) \right], \qquad (1)$$

where $\Delta F$ is the hom. nucleation barrier, $n_c$ is the hom. critical nucleus size, and $f_V(\theta) = V_{het}/V_{hom} = (1 - \cos \theta)^2 (2 + \cos \theta)/4 \in [0, 1]$ is the volumetric shape factor. The latter describes the enhancement that is achieved by nucleating on top of a flat substrate with a certain contact angle $\theta$. The nucleus shape in the het. case is a spherical cap of volume $V_{het}$ rather than a full sphere of volume $V_{hom}$ as in the hom. case. For illustrative purposes, some examples of different free energy profiles and sketches of the corresponding contact angles are shown in Fig. 1. If line-tension effects are negligible, $f_V$ is independent of the nucleus size $n$[1] and we can express $f_V$ in terms of the het. and hom. free energies:

$$(f_V)^{1/3} = \frac{F_{het}(n)}{F_{hom}(n)} \left[ 1 - \frac{2}{3} \left( \frac{n}{n_c} \right)^{1/3} \right] + \frac{2}{3} \left( \frac{n}{n_c} \right)^{1/3}. \qquad (2)$$

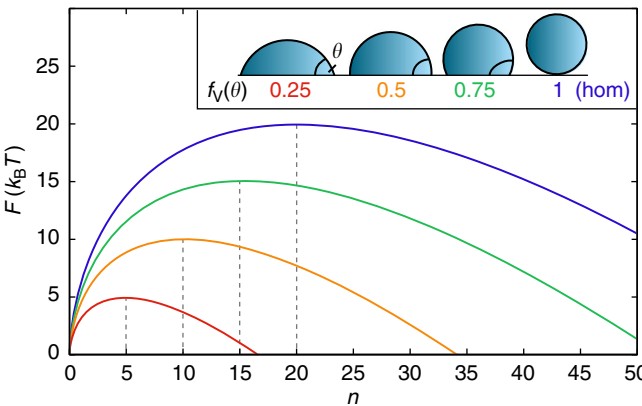

**Fig. 1** The traditional view of free energy profiles in hetCNT. Free energy profiles are shown for a crystalline cluster containing $n$ molecules for different contact angles. The inset shows the definition of the contact angle and representative droplets corresponding to different values of $f_V(\theta)$

Interestingly, Eq. (2) requires only knowledge of the hom. $n_c$, as the free energies $F(n)$ can be obtained for arbitrary cluster sizes $n$. Knowing the value of $f_V$ for a given substrate is fundamental as it encodes all information about the nucleation enhancement, which is reflected in the fact that all the curves in Fig. 1 retain the same functional shape and the steepness ratio

$$\chi(f_V) = \frac{\Delta F(f_V)}{n_c(f_V)} = \frac{f_V \cdot \Delta F_{hom}}{f_V \cdot n_{c,hom}} = \frac{\Delta F_{hom}}{n_{c,hom}} = \chi, \qquad (3)$$

is independent of the enhancement. This has been key in several nucleation studies[32–34], on evaluating the performance of CNT[35,36], or as a bridge between atmospheric cloud models and the microscopic description of ice nucleation[37,38]. It is however tremendously challenging to measure $f_V$, since knowledge of difficult-to-obtain quantities like $\Delta F$ is needed. In principle, one has easier access to precritical quantities, such as the probability $P(n)$ of finding a cluster of size $n < n_c$ in the liquid state. Since $P_{het}(n)/P_{hom}(n) \propto F_{het}(n)/F_{hom}(n)$, Eq. (2) could be evaluated from the statistics of precritical clusters (termed precritical fluctuations) without having to observe the rare nucleation event itself. Although many aspects of nucleation have been studied in great detail, the role of precritical fluctuations in het. nucleation is less well understood. A deeper understanding could potentially be exploited to gain insight into fundamental aspects of het. crystal nucleation.

In this work, we aim to understand precritical fluctuations and their connection to nucleation kinetics by comparing cluster fluctuations on two model substrates that enhance the nucleation of ice to the same extent. To this end we perform an extensive set of molecular dynamics simulations. From these it emerges that the traditional hetCNT picture can break down, because a substrate can facilitate the formation of different polymorphs. As a consequence, when using hetCNT, one must choose a different bulk-reference to describe the nucleation process correctly. Although here we illustrate the potential role of precritical fluctuations in the context of CNT, in principle they could be used with any theory that provides a free energy profile for nucleation. We hope that the new insight obtained furthers the theoretical understanding of het. nucleation, polymorph selection, and the role of precritical fluctuations.

## Results

**Identical rates despite different precritical fluctuations.** The results for the ice nucleation rates of supercooled water in contact with two model substrates (termed s1 and s2) are summarized in Table 1. Since nucleation rates can differ by many orders of magnitude, we can label the resulting rates from our systems as essentially identical. The presence of the substrates compared to the hom. case increases the nucleation rate by many orders of magnitude, since no freezing is observed for hom. simulations at 218 K or even 210 K. Based on the rates of Li et al. for hom. nucleation[39], we estimated the enhancement to be between 5 and 8 orders of magnitude. In the Supplementary Note 1 we provide a

**Table 1 Computed nucleation rates $J$ for the two systems s1 and s2 at two temperatures: For the sake of comparison we have normalized the rates by the water-substrate contact area**

| System | s1 | s2 |
|---|---|---|
| $J_{218K}$ (ns⁻¹ Å⁻²) × 10⁻⁶ | (7.18 ± 1.30) | (4.23 ± 0.71) |
| $J_{221K}$ (ns⁻¹ Å⁻²) × 10⁻⁷ | (2.82 ± 1.02) | (2.88 ± 0.77) |

The error in the rate has been estimated by Jackknife resampling

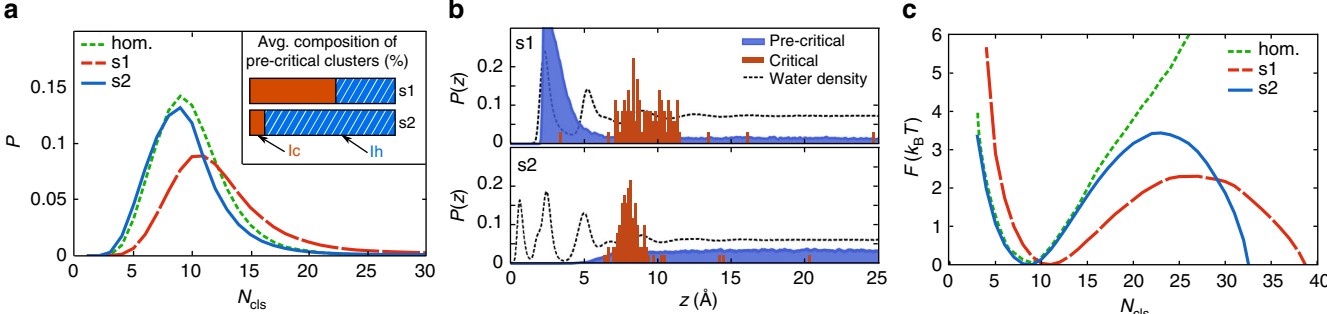

**Fig. 2** Cluster formation and free energy of ice formation for the two substrates at 218 K. **a** Probability distribution of the size of the biggest ice-like cluster $N_{cls}$. The inset shows the average composition of precritical clusters within 5 Å of the surface. **b** Probability distribution (blue) for the z component of the center of mass of ~100,000 precritical clusters. Positions of critical clusters are indicated in red. The water density is also indicated by dashed black lines (arbitrary scale). **c** Free energy profiles as a function of $N_{cls}$

brief assessment of why the substrates enhance the nucleation process. Now, however, we focus on the precritical fluctuations in s1, s2, and the hom. case . In Fig. 2a we plot the size distribution of the biggest ice-like cluster that can be found in each snapshot from the trajectories. The probability densities for the hom. system and s2 line up almost exactly, while the probability of spontaneously forming large clusters is clearly enhanced in s1. Furthermore, we plot the spatial probability distribution of these precritical clusters in Fig. 2b. For s1, there is clearly a large peak near the surface. In contrast, the same density for s2 is nearly flat, which means—contrary to the expectations of hetCNT—fluctuations are not enhanced near the substrate. We checked that no persistent structures are detected as precritical clusters, for example, confirming that the peak for s1 stems from small ice-like patches that are continuously fluctuating rather than from a permanent ice-like overlayer. The difference between the two systems is also reflected in the asphericity of the nuclei formed, where s1 tends to form flatter clusters than s2 (Supplementary Note 2). We verified that these findings hold for the same order parameter with the first hydration shell included in the cluster-definition, as well as for a different choice of underlying spherical harmonics (Supplementary Note 3). The difference in precritical fluctuations is surprising as the nucleation rate in both systems is effectively the same. In the framework of hetCNT this would imply that both substrates should have the same free energy profile and steepness ratio $\chi$ (Eq. (3)), which is incompatible with the differences in precritical cluster sizes we observe. This shows that precritical fluctuations and hetCNT, as commonly applied, do not adequately describe the behavior observed on the two surfaces studied.

**Different polymorphs yield different free energy profiles**. To understand why the fluctuations are so different on the two substrates, we now look at the free energy profiles for nucleation. These are shown in Fig. 2c, where it can be seen that the curve for s2 follows the hom. one more closely than s1 does. The fact that the curves for s2 and hom. are similar is consistent with the fact that we find essentially the same cluster size distribution in the two. The free energy profile for s1 is flatter, and therefore, the formation of larger clusters is less costly compared to s2 and hom. It is also clear that in disagreement with hetCNT for the ratios we find $\chi_{s1} \neq \chi_{s2}$ (see Eq. (3)). The results for 221 K can be found in the Supplementary Note 4 and show essentially the same features. Furthermore, a reconstruction of $F$ as a function of the CNT coordinate $n$ reveals similar differences, particularly in the steepness of the free energy paths. If hetCNT was adequate for our observation the functional shape of s1 and s2 should be

identical (as illustrated in Fig. 1). But for instance, $\Delta F$ and $n_c$ are not being scaled by a single factor, which suggests that at least one additional degree of freedom might enter in het. nucleation. In what follows, we show that the polymorph, which can be influenced by the substrate, can account for this.

We try to understand why we see different functional forms for the free energy curves. In line with other studies[36,40,41], we find that for s1 stacking-disordered ice $I_{sd}$ has formed, where the basal face of ice is in contact with the substrate. This is also the type of ice that forms in hom. simulations[19,39,42,43]. The shape of the free energy profile for s1 and the enhanced precritical fluctuations compared to the hom. case can thus be explained by traditional hetCNT. In contrast, s2 forms a crystal face (primary prism) in contact with the surface that is only found in hexagonal ice $I_h$, but not in $I_c$. Thus, $I_c$ layers cannot grow on top of that and subsequently the stacking disorder, which usually leads to the formation of $I_{sd}$, is strongly disfavored at the surface (illustrated in Fig. 3d). The traditional hetCNT does not work for that because the nucleus on s2 is purely hexagonal, and therefore, not related to the hom. one that traditional hetCNT chooses as reference (stacking-disordered). Hence, it is no surprise that the shape of the free energy profile for s2 is different from s1, despite their identical nucleation rates. The precritical fluctuations in s2 appear unaltered compared to the hom. case, not because the substrate has no impact, but rather because they are fluctuations of a different polytype and this comparison is ill-defined. This can be seen in the inset of Fig. 2a, where we show that an average precritical cluster within 5 Å of the surface in s1 is stacking disordered with a cubicity (fraction of $I_c$) of ~60%, while s2 forms 90% pure $I_h$ clusters. The apparent ~10% of $I_c$-like molecules in s2 are due to uncertainties in classifying interfacial molecules at the edge of the cluster. We have visually verified that, in contrast to s1, in s2 we never observed clusters near the substrate that are in their core stacking disordered (see also Fig. 3c). Although the statistics we have for critical clusters are worse, we note that on each surface their composition was nearly identical to the one of the respective precritical clusters. This further suggests that there is a causal connection between critical and precritical clusters.

**The polymorphic differences remain at higher temperature**. To understand if our findings hold at higher temperatures, we performed metadynamics simulations in our two systems at an elevated temperature of 235 K. This is around the highest temperature we can aim to study with our system size (as we expect the hom. critical cluster size to be ≈ 600 molecules[41]). In Fig. 3a we show the free energy profiles obtained, where we note that the variable s describes the path from a liquid ($s ≈ 1.1$) to a

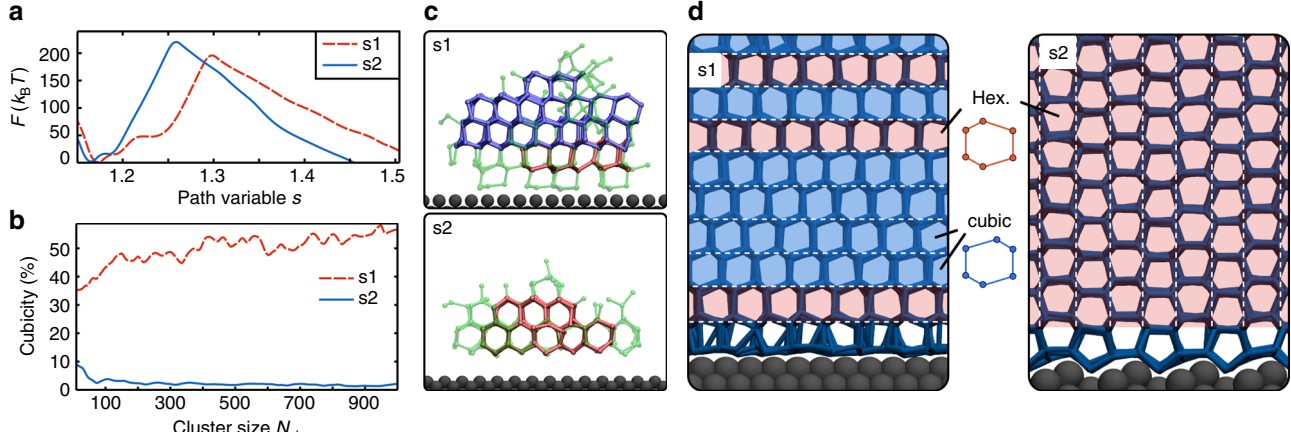

**Fig. 3** Metadynamics results for heterogeneous ice nucleation at 235 K. **a** Free energy profiles as a function of the path variable $s$ that describes the progression from liquid at $s \approx 1.1$ toward the soft-wall at $s = 1.5$. **b** Cubicity as a function of the cluster size $N_{cls}$. **c** Representative snapshots of critical clusters. Ice molecules and bonds are transparent green, while double diamond cages are blue, and hexagonal cages are red. **d** Subsection of the fully frozen cells, illustrating the substrate-induced polytype selection by avoiding the stacking disorder

frozen ($s \approx 1.9$) simulation cell. We have employed an artificial soft-wall at $s = 1.5$ to aid convergence for the region describing cluster sizes relevant to nucleation rather than growth. From these simulations we obtain (details in the Supplementary Methods 1) a free energy barrier on s1 of $204 \pm 5\ k_B T$ and s2 of $227 \pm 5\ k_B T$, and critical cluster size on s1 of $211 \pm 11$ and s2 of $104 \pm 3$. Finding that $\Delta F_{s1} < \Delta F_{s2}$ and $n_{c,s1} > n_{c,s2}$ is entirely consistent with the trends obtained at lower temperatures. In addition, it can be seen from Fig. 3b that the polytype of ice formed in s1 and s2 is not the same, the former being $\approx 55\%$ stacking-disordered and the latter being almost purely hexagonal. The deviations for smaller clusters are once again artifacts of the local order parameter employed at the cluster interface, where the classification is ambiguous. To illustrate the difference in the cluster cores, Fig. 3c shows the representative snapshots for critical clusters in s1 and s2, and highlights hexagonal and double diamond cages, the building blocks of $I_h$ and $I_c$[43] that are a stronger topological feature than the local order parameter. In panel (**d**) of Fig. 3 we illustrate that the substrate in s2 avoids the stacking-disorder by stacking ice double-layers perpendicular to the surface, which is a result of the crystal face (prism) in contact with the surface. We note in passing that this could be a general recipe for water and other tetrahedral liquids (e.g., group-IV elements or silica) and could also be exploited to design surfaces that nucleate pure cubic ice. Overall, the findings for the higher temperature agree with the simulations at lower temperature, suggesting that our reasoning also holds for situations where precritical and critical clusters are separated by more than one order of magnitude in size.

## Discussion

We try to place the results of this study in a broader context and discuss some of the implications of our findings. The first consequence drawn from the possible occurrence of different polymorphs is that the fundamental result of hetCNT that reads $n_{c,het} = f_V \cdot n_{c,hom}$ and $\Delta F_{het} = f_V \cdot \Delta F_{hom}$ is not true for cases where the substrate promotes the formation of a polymorph different than the one that is formed homogeneously. This is because the enhancement factor $f_V$ is only properly defined if the het. quantity it describes refers to the hom. reference of that polymorph. In general, when thinking about het. nucleation there are three possible ways to account for the enhancement factor: (i) an expression in terms of a shape factor is $f_V = V_{het}/V_{hom}$;

(ii) an expression in terms of a nucleus factor is $f_N = n_{c,het}/n_{c,hom}$; and (iii) an expression in terms of a potency factor is $f_P = \Delta F_{het}/\Delta F_{hom}$. These three definitions are equivalent under the assumption that they describe events where the same polymorph has been formed. However, if different polymorphs are compared the concept of the enhancement factor becomes ill-defined. We derive in the Supplementary Note 5 correction factors in the framework of hetCNT that account for this change. The fact that increasing the temperature accentuated the difference in the free energy profiles observed on the two substrates is an indication that effects like line-tension[44] and cluster asphericity[12] are not the main reason for our observation (as those likely decrease with increasing temperature/increasing cluster sizes), but rather it is caused by the different polymorphs. Hence, we believe that the polymorph is a separate issue that should be taken into account in a comprehensive (het.) nucleation theory, in addition to known shortcomings of CNT or its het. extension. We speculate that for the same reason the polymorph could even be the most relevant deviation from hetCNT at high temperatures.

Another implication of this work is that precritical fluctuations are comparable for different substrates only if compared to the correct hom. fluctuations of their corresponding polymorph. In our study, the comparison of the precritical fluctuations of s1 and s2 with the hom. case would have resulted in the conclusion that s1 enhances the nucleation and s2 does not (Fig. 2a), while they actually lead to nearly identical enhancement. We have illustrated this in Fig. 4a, b, with hom. and het. nucleation profiles for two different polymorphs. The gray-shaded area and its reach on the $x$-axis illustrates what cluster sizes can be reached through thermal fluctuations. This ultimately determines the extent of precritical fluctuations and is very different for the two polymorphs as a result of their different hom. free energy profiles. Upon comparing to the hom. nucleation of a single (homogeneously dominant) polymorph (which without loss of generality we assume to be hom,1 in Fig. 4c) the apparent discrepancy becomes clear.

In summary, we have presented a comparative study of molecular dynamics simulations of het. ice nucleation on two distinct model substrates. It was shown that, in disagreement with hetCNT as traditionally applied, their precritical fluctuations can differ substantially, and yet identical nucleation rates are obtained, which we attribute to the formation of different polytypes. From this we draw the following conclusions:

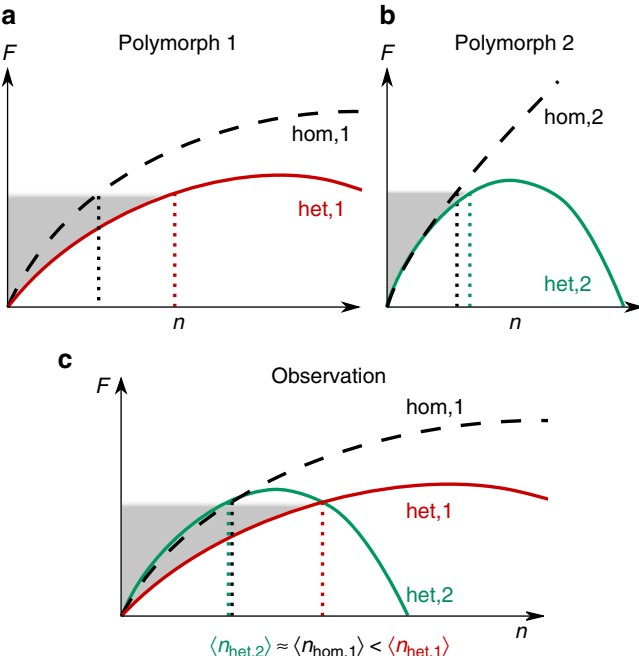

**Fig. 4** Schematic illustration of the connection between free energy profiles, precritical fluctuations, and polymorph. **a**, **b** Resulting het. free energy profiles for two different polymorphs, which belong to the same functional family as their hom. reference. The extent of thermal fluctuations is indicated by the gray-shaded area. **c** Observation in a simulation or experiment where the hom. nucleation of the dominant polymorph 1 is compared to het. nucleation events that form the same (het,1) and a different (het,2) polymorph. While for the het,1 profile the critical nucleus and the barrier are scaled by a same factor (as predicted by hetCNT), these are scaled differently for the het,2 profile when compared to hom,1. Note that the graphs are qualitatively equivalent to our simulation results, but do not result from a fit or parametrized model

1. Substrates can promote the formation of metastable phases by templating crystal faces that are unique to the respective polymorph. This is an extension of the rationale applied in experimental studies where iso-structural templates are used[45,46], since the substrate does not require the same structure, but rather any structure that nucleates the right crystal face. In particular, for materials with different stackings (e.g., ice, group-IV elements or silicates), the templating of faces so that the stacking direction is perpendicular to the surface normal seems most promising and could potentially avoid stacking-disorder.

2. Traditionally applied hetCNT can break down when a polymorph nucleates, which is not the dominant homogeneous polymorph. This should be corrected by choosing the right bulk-reference, an aspect that is largely disregarded in the nucleation literature.

3. We hypothesize that the extent of precritical fluctuations carries information about the enhancement of nucleation and can serve as an early indicator of which polymorph will form. This provides a possible route to efficiently rank the nucleation ability of substrates and will be the subject of future investigation. Since precritical fluctuations are less sensitive to finite size effects than approaches in which the full nucleation path is examined, they could prove particularly useful for studying nucleation at very-low levels of supercooling, provided one pays attention to the comparability of systems (same contact area, temperature etc.).

**Table 2 Summary of simulation parameters such as the lattice constant $a_{fcc}$ of the substrate and the water-substrate interaction parameters $\epsilon_{ws}$, $\sigma_{ws}$**

| System | Surface | $a_{fcc}$ (Å) | $\epsilon_{ws}$ (kcal mol$^{-1}$) | $\sigma_{ws}$ (Å) |
|---|---|---|---|---|
| s1 | fcc(100) | 3.649 | 0.43 | 2.488 |
| s2 | fcc(211) | 4.158 | 0.48 | 2.582 |

While these implications have arisen from simulations of het. ice nucleation, it is clear that they are general to the phenomenon of het. nucleation and particularly relevant to the description of materials that display polymorphism, such as e.g. alumina[47], silicon[46], xenon[48], n-alkane[49], or the epilepsy drug carbamazepine[45]. A quantitative treatment of nucleation in these systems such as comparing the enhancement of nucleation to the hom. case requires information about the polymorph that is formed. If the polymorph is disregarded this could lead to false inferences about nucleation rates, mechanisms, or the accuracy of CNT and might also cause widely used CNT-based models that use a single enhancement factor $f_V$ as a parameter to appear inadequate. Further studies aimed at understanding and potentially exploiting precritical fluctuations are needed.

## Methods

**Unbiased molecular dynamics**. We performed molecular dynamics simulations of het. ice nucleation with 18,000 water molecules, represented by the coarse-grained mW model[50], a model that is widely used to study phenomena involving water[17,19,36,40,41,51–54]. The water molecules are placed in a film geometry on top of two pristine, rigid fcc surfaces (termed s1 and s2). See the Supplementary Methods 2 for the explicit structure of each surface and additional computational details. These surfaces have proven useful in disentangling the contributions of lattice match and hydrophobicity to het. ice nucleation[24]. The two examples presented here have been selected because of their striking difference in precritical fluctuations. While not aiming at representing any specific material, they are most similar to metal–water interfaces[55,56]. The substrate–water interaction is given by a Lennard–Jones potential tuned to achieve the same absolute nucleation rate (see Table 2 for interaction parameters). Following established protocols[24,57], we first equilibrate each structure for 10 ns at 300 K. Then production runs are quenched to the target temperature and coupled to a 10-fold Nosé–Hoover chain[58] to sample the NVT ensemble, integrating the equations of motion with a timestep of 10 fs. The relaxation time after the quench is on the order of 10 ps, and can thus be considered non-disturbing to the nucleation. The nucleation events themselves are detected by a sudden drop in the potential energy, upon which we terminate the computation and collect the current time as induction time. A total of 100 simulations for each of the two substrates at 218 K and 50 simulations at 221 K have been performed with large-scale atomic/molecular massively parallel simulator[59]. From the collection of induction times we fit the survival probability as

$$P_{sur} = \exp[-(J \cdot t)^\gamma] \qquad (4)$$

to obtain the nucleation rate $J$, where $\gamma$ is a correction factor that accounts for possible non-exponential kinetics.

**Free energy reconstruction from unbiased MD**. For the reconstruction of free energy we made use of the kinetic reconstruction method[60]. To this end we computed the steady-state distribution $P(x=n)$ of the order parameter $x$ and the mean first passage time $\tau(x=n)$ that it on average takes for the system to reach the state $x=n$. The free energy can be reconstructed from this via:

$$\beta F(x) = C + \ln[B(x)] - \int_a^x \frac{dx'}{B(x')}, \qquad (5)$$

where $C$ is an arbitrary constant, $B(x)$ is a help quantity defined by

$$B(x) = \frac{1}{P(x)}\left[\int_a^x P(x')dx' - \frac{\tau(x)}{\tau(b)}\right], \qquad (6)$$

and $a$ and $b$ are boundary conditions. The choice of the latter as well as the numerical integration method did not meaningfully alter any of the results.

**Identification if ice**. Ice-like molecules were detected using an order parameter according to Li et al.[39] as implemented in PLUgin for MEtaDynamics 2[61,62]. First

we compute for each molecule $i$ the quantity $q_{lm}(i)$ as follows:

$$q_{lm}(i) = \frac{1}{N_b(i)} \sum_{k=1}^{N_b(i)} Y_{lm}(\theta_{ik}, \phi_{ik}), \quad (7)$$

where, the sum goes over the $N_b(i)$ neighbors of molecule $i$, $Y_{lm}$ are spherical harmonics, and $\theta_{ik}$ and $\phi_{ik}$ are the relative orientational angles between the molecules $i$ and $k$. For a given $l$ we compute the quantity for all possible values of $m$ and store them in a vector $\vec{q}_l(i)$ containing $2l + 1$ components. Finally, we calculate values of $q_l$ according to:

$$q_l(i) = \frac{1}{N_b(i)} \sum_{k=1}^{N_b(i)} \frac{\vec{q}_l(i) \cdot \vec{q}_l(k)}{|\vec{q}_l(i)| \cdot |\vec{q}_l(k)|}. \quad (8)$$

For the particular choice of $l = 3$ the values of $q_3$ can distinguish both between the solid and liquid molecules as well as between cubic and hexagonal ice. For values of $q_3 < -0.69$ we classify the molecule as ice-like. Additionally, if $q_3 < -0.85$ the molecule belongs to cubic ice and otherwise to hexagonal ice (the distribution of $q_3$ for different water phases can be found elsewhere[39]).

**Metadynamics simulations.** We performed well-tempered metadynamics simulations[63,64] with 20 walkers[65] at 235 K, and employed smaller simulation boxes with 8000 molecules in a $60 \times 60 \times 78$ Å cell with 3D periodic boundary conditions. Nucleation was facilitated by biasing the path variables[66] $s$ and $z$ constructed by measuring the generalized distances of the systems permutation invariant vector[67,68] (PIV) to two reference states (liquid and frozen simulation cells obtained from brute-force MD at 205 K). In essence, the PIV is the vector of irreducible adjacency matrix entries ordered by magnitude, where entries decay smoothly from one to zero for intermolecular distances beyond 3.4 Å. The metadynamics parameters were Gaussian height $\delta = 0.2$ kcal/mol, Gaussian width $(\sigma_s, \sigma_z) = (0.022, 0.38)$, deposition stride 2 ps, and a bias factor of 50. We employ a repulsive wall at $s = 1.5$ to restrain our simulation to cluster sizes relevant to nucleation. The resulting free energy profiles were checked for convergence by reweighting[69] to the one-dimensional free energy profile $F(s)$ (Supplementary Methods 1). The critical cluster sizes were obtained via a committor analysis[70] seeded from the metadynamics trajectories.

**Data availability.** The data that support the findings of this study are available from the corresponding author upon reasonable request.

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

## Acknowledgments

We thank T. Li, G. Tribello and S. J. Cox for comments and suggestions. This work was supported by the European Research Council under the European Union's Seventh Framework Programme (FP/2007–2013)/ERC Grant Agreement No. 616121 (HeteroIce project). A.M. is supported by the Royal Society through a Royal Society Wolfson Research Merit Award. We are grateful for computational resources provided by the London Center for Nanotechnology, the UCL Grace High Performance Computing Facility (Grace@UCL), the Materials Chemistry Consortium through the EPSRC Grant No. EP/L000202 and the UK Materials and Molecular Modeling Hub for computational resources, which is partially funded by EPSRC (EP/P020194/1). This work was supported by French state funds managed by the ANR within the Investissements d'Avenir programme under reference ANR-11-IDEX-0004-02, within the framework of the cluster of excellence MATériaux Interfaces Surfaces Environnement (MATISSE) led by Sorbonne Universités.

## Author contributions

M.F., G.C.S., and A.M. conceived the research. M.F. performed the simulations and analysis. F.P. and S.P. contributed to the metadynamics simulations and analysis. M. F., G.C.S., and A.M. interpreted the results and wrote the manuscript.

## Additional information

**Competing interests:** The authors declare no competing financial interests.

