## [Peer Review File · Nature Communications]

Reviewer #2 (Remarks to the Author):

The manuscript by Fitner et al presents a very interesting simulation study of heterogeneous crystal nucleation of water. The main novel aspect of this work is the analysis of pre-critical crystalline clusters that form favored by the presence of surfaces with different characteristics. The authors show evidence that the two different substrates analyzed stimulate the formation of different pre-critical crystal polymorphs, which in turn are crucial to decide the rate and fate of the final crystal formed. In author's view, this possibility of forming different polymorphs is not accounted for by classical heterogeneous nucleation theory (hetCNT), and a correction factor is introduced in order to patch it.

The topic of the work is of obvious impact and broad interest in atmospheric, pharmaceutical, biological and material science. The work is of high-quality and nicely written. However, I have several doubts and important concerns that I would like the authors to clarify before publication can be recommended, in order to dissipate doubts about the relevance of the results and to support more soundly their conclusions. More specifically:

- One of my main concerns is the fact that the simulations have been done well below the homogeneous freezing temperature. Such large undercoolings are of course required to observe spontaneous crystallization in a brute-force simulation, even in the heterogeneous case. However, low temperatures correspond to small nucleation barriers and relatively small critical cluster sizes. Both features could overemphasize the importance of pre-critical fluctuations in the observed crystallization process. The relative stability of polymorphs may strongly depend on cluster size (in a similar way to what happens to the relative stability of stacking-disorder ice vs hexagonal ice in small vs macroscopic crystals). Would the same conclusions hold for more realistic undercoolings? The results in the SI for $T=221$ K seem to suggest a tendency to decrease the differences in the free energy profiles between the two substrates as the temperature is raised. I am aware that it is computationally hard to do another simulation at a higher temperature, but in this case I think that it will be necessary to show solid evidences that for (more realistic) higher T and significantly larger critical cluster sizes, the effect of different pre-critical polymorphs is still relevant. It would be crucial to see if the composition, size and free energy of formation of the critical cluster tends to be the same for both substrates as the critical cluster size gets larger. If that is the case, the small free energy differences in the formation of different pre-critical polymorphs will not be relevant for the final rate and outcome of the freezing process at realistic undercoolings.
- In the interpretation of the results in connection with hetCNT, there are some weak points that need to be properly addressed. First, the authors completely ignore line tension effects, but this is not at all justified in the formation of such small clusters at large undercoolings. Small nucleation barriers can be dominated by line tension effects (see for instance Ref. 20). HetCNT taking into account line tension yields a functional form of the free energy landscape substantially different from Eqs. 1 and 2, and that may not retain the same functional shape. In addition, for small wetting angles, a pancake cluster would probably be more likely than a spherical cap (as mentioned by the authors in the main text and in the SI). The free energy expression for such a shape is different from Eqs. 1 and 2. Both effects (i.e. different line tensions and pancake-shaped crystals) may be dominant and could make unnecessary and meaningless the extension of CNT suggested by the authors.

Regarding the main conclusions:

- The first paragraph is a bit confusing. The authors emphasize the fact that different pre-critical fluctuations lead to similar nucleation rates. But in the Methods, they clearly state "the substrate-water interaction... was tuned to achieve the same absolute nucleation rate". Since the substrate roughness, lattice spacing, and Lennard-Jones parameters of interaction with water are different, it is not surprising that interfacial energies, line tensions, and wetting angles (if they can be somehow defined for such small crystals) would be different and would lead to the preferential formation of small clusters with different structures.

- The first conclusion may be reinforced by the results of this work, but it is not new. Moreover, it is not fully demonstrated that the structure of the most abundant pre-critical polymorph is eventually the same as the one in the critical or larger clusters, for larger nucleation barriers and more realistic undercoolings.
- The second conclusion is not evident that holds for any undercooling and critical cluster size. There is the possibility that line tension and the non-spherical cap shape of the clusters will be more relevant than the differences between the bulk properties of the different polymorphs.
- The third one is not well substantiated. The most probable largest pre-critical crystal cluster is strongly affected by finite size effects, and depends crucially on the total number of molecules N used in the simulation and the temperature (the larger the N and the smaller the temperature, the larger will be the size of the most probable largest cluster). In addition, the different probabilities of having different polymorphs for small fluctuations might be irrelevant in the formation of the critical cluster, for low (and more realistic) undercoolings and correspondingly large critical clusters.

Finally, some minor issues:

- Eq. 6 is missing a minus sign, and the need of the parameter γ is unclear; in a true nucleation (activated) process, the survival probability is exponential.
- Since the mean first passage time seems to be evaluated in the reconstruction of the free energy, why was not used to get an estimate of the values of nucleation rates?
- It would be helpful to report also the structure or composition of the critical clusters, as it has been done for the pre-critical clusters in Fig. 2a.
- The asphericity of clusters may strongly depend on their size. It would be also interesting to have a plot, similar to Fig. S3, but in terms of cluster size rather than distance from the substrate.

In summary, although the ideas and results presented in this work are potentially interesting, there are many loose ends that the authors need to address doing a major revision before the paper can be recommended for publication.

Reviewer #3 (Remarks to the Author):

This manuscript reports what is a rather interesting study of factors important to heterogeneous nucleation. Employing molecular dynamics simulations, the authors investigate the influence of two model surfaces on the nucleation of ice, where the surfaces have been parameterized to give essentially the same enhancement of the nucleation rate (relative to bulk). The authors then demonstrate that the pre-critical (structural) fluctuations are notably different for each system and that heterogeneous classical nucleation theory (hetCNT) is unable to account for this behavior. This work thus provides important insights into the possible connections between specific structures (appearing prior to the nucleation of the system) and the thermodynamics and kinetics of crystal nucleation; as such it is potentially of great interest and broad implications. The results of this manuscript lead the authors to identify two primary conclusions (where each is discussed in its own section). The first of these is that the formation of different polymorphs leads to different heterogeneous free energy profiles. I generally found this section well supported and an interesting read. However, I found the second section, which describes an extension to CNT to account for the observed differences, much less convincing. Consequently, I would recommend that the authors either scale-back considerably their second section (on the Extension of CNT), or that they significantly improve their justification of this approach and more clearly describe its assumptions/limitations (further details are provided below). Hence, I would support the eventual publication of this work once the authors have addressed this major concern, along with the other issues I have detailed below.

1) I found the section discussing the Extension of CNT to be the weakest part of this manuscript – hence my recommendation is that it either be appropriately justified, or be made a simpler, more phenomenology discussion (I would tend to favor the latter).

- a) New relationships for describing heterogeneous nucleation are introduced on page 7. However, the authors do a poor job of outlining clearly the assumptions on which these expressions are based (although afterwards, statements such as “provided the approximations of CNT as reasonable” appear).
- b) These relationships assume that properties of the bulk phases (i.e. chemical potentials and surface free energies) remain valid when applied to crystal nuclei. In the present case, the two systems being studied exhibit critical nuclei consisting of about 20-30 water molecules. I might hope that the authors suspect that there are major limitations in assuming these macroscopic properties are applicable to such small (nanoscale) clusters. Indeed, given that the critical nucleus for the corresponding homogeneous process is considerably larger (hence more likely to be reasonably described by macroscopic parameters), perhaps the breakdown of these assumptions (which become unique to each system) is also a major part of the observed phenomenology?
- c) In view of the above, it is then crucial for the authors to demonstrate that their new relationships (on page 7) can be used to describe (fit) the behavior of the current systems.
- i) It is not obvious how these relationships were used to arrive at the sketch provided in Figure 3 – specifically what assumptions were made for the appropriate parameter values?
- ii) Reasonable estimates for the difference in the free energies of the two (bulk) ice phases are known (see for example theoretical predictions by Molerino and co-workers, or estimates from experiment by Jahari). Additionally, since the ice phases differ only in their stacking (of hexagonal layers), we would expect their (bulk) densities to be essentially identical. It should then be possible to use these values to test the validity of their expressions.

2) Some additional minor issues:

- a) On page 3, second to last sentence of the first paragraph: I found the wording here confusing.
- b) On page 3, last sentence of the first paragraph: claiming “have never been studied” is perhaps a bit of an overstatement here. Others have certainly looked at structural fluctuations prior to nucleation (e.g., for ice, consider the work of Debenedetti – ref. 47, and for gas hydrates the work of the groups of Kusalik and of Molinero). What is important in the current work is the characterization and the degree of distinction. Appropriate clarification should be added.
- c) Page 5: the authors use the terms “identical” (line 2) and “the same” (second line from bottom), when I think they mean “essentially identical” and “essentially the same”.
- c) On page 5, line 8: the term “of each frame” should be clarified.
- d) On page 6, line 7: the authors state “suggests an additional degree of freedom” when it would be more correct to state “suggests that at least one additional degree of freedom”. (There are likely deeper issues associated with choices of order parameters here, but such a line of inquiry would be beyond the scope of this work.)
- e) On page 6, second paragraph: in this paragraph, the authors indicate that formation of Ic layers is “strongly disfavored” for the s2 surface, yet at the end of the paragraph they report that 90% Ih clusters (which implies 10 % Ic clusters). Apparently the level of selectivity of the structural fluctuations appears not to be extremely selective (rather there seems to be enrichment of Ih clusters). Either the authors should adjust their language, or explain the discrepancy.
- f) On page 7, second line after Eq. (5): “This explains why ...” is not appropriate here (since other explanations are possible). More appropriate to state “This may account for why ...”
- g) Sentence beginning at the bottom of page 7: I found this sentence rather confusing (in part because of the use of the word “definitions”). I would suggest rewording (e.g. “In general when thinking about het. nucleation there are three possible ways to account for the enhancement factor: (i) through an expression in terms ...”)
- h) Reference 48 should be removed, or a publication provided.

Reviewer #2 (Remarks to the Author):

The manuscript by Fitner et al presents a very interesting simulation study of heterogeneous crystal nucleation of water. The main novel aspect of this work is the analysis of pre-critical crystalline clusters that form favored by the presence of surfaces with different characteristics. The authors show evidence that the two different substrates analyzed stimulate the formation of different pre-critical crystal polymorphs, which in turn are crucial to decide the rate and fate of the final crystal formed. In author's view, this possibility of forming different polymorphs is not accounted for by classical heterogeneous nucleation theory (hetCNT), and a correction factor is introduced in order to patch it.

The topic of the work is of obvious impact and broad interest in atmospheric, pharmaceutical, biological and material science. The work is of high-quality and nicely written. However, I have several doubts and important concerns that I would like the authors to clarify before publication can be recommended, in order to dissipate doubts about the relevance of the results and to support more soundly their conclusions.

We thank the reviewer for this assessment of the relevance of our work and for carefully reading the manuscript. We address their concerns with the responses outlined below. Most importantly, we add new results from metadynamics simulations on our systems at much higher temperature which reinforce our initial conclusions.

More specifically:

- One of my main concerns is the fact that the simulations have been done well below the homogeneous freezing temperature. Such large undercoolings are of course required to observe spontaneous crystallization in a brute-force simulation, even in the heterogeneous case. However, low temperatures correspond to small nucleation barriers and relatively small critical cluster sizes. Both features could overemphasize the importance of pre-critical fluctuations in the observed crystallization process. The relative stability of polymorphs may strongly depend on cluster size (in a similar way to what happens to the relative stability of stacking-disorder ice vs hexagonal ice in small vs macroscopic crystals). Would the same conclusions hold for more realistic undercoolings? The results in the SI for $T=221$ K seem to suggest a tendency to decrease the differences in the free energy profiles between the two substrates as the temperature is raised. I am aware that it is computationally hard to do another simulation at a higher temperature, but in this case I think that it will be necessary to show solid evidences that for (more realistic) higher T and significantly larger critical cluster sizes, the effect of different pre-critical polymorphs is still relevant. It would be crucial to see if the composition, size and free energy of formation of the critical cluster tends to be the same for both substrates as the critical cluster size gets larger. If that is the case, the small free energy differences in the formation of different pre-critical polymorphs will not be relevant for the final rate and outcome of the freezing process at realistic undercoolings.

The point that is made by the referee is very important and we agree that the strong supercooling (that is needed for a brute-force study) is the main weakness of our work. For this reason we decided to perform new simulations at a higher temperature, combining well-tempered metadynamics with a relatively new order parameter, called permutation invariant vector (PIV) as presented in the recent literature (arXiv:1703.00753). These simulations were done at a temperature of 235K which is near the upper limit where we can expect simulations of usual size ($\sim 10,000$ water molecules) to be free of finite size effects (the expected hom. critical cluster size for mW at 235K is ~ 600). Furthermore, significantly higher

temperatures can become much too costly to study, even with enhanced sampling techniques.

Our results are summarized in a new display figure (figure 3 in the revised manuscript):

Caption: Metadynamics results for heterogeneous ice nucleation at 235 K: a) Free energy profiles as a function of the path variable s that describes the progression from liquid at $s \approx 1.1$ towards the soft-wall at $s = 1.5$. b) Cubicity as a function of the cluster size N_{cls} . c) Representative snapshots of critical clusters. Ice molecules and bonds are transparent green while double diamond cages are blue and hexagonal cages are red. d) Subsection of the fully frozen cells, illustrating the substrate-induced polytype selection by avoiding the stacking disorder.

In there we show that

1. The free energy profiles (panel a) of the two systems are very different and do not approach each other as queried by the referee. The trends regarding the free energy barriers ($F_{s1} < F_{s2}$) and critical cluster sizes ($n_{c,s1} > n_{c,s2}$, obtained from committor analysis) are precisely as for the lower temperatures, not too say much stronger.
2. The polytype of ice that is found in the two systems is drastically different, for both the pre-critical and critical clusters, see panel b). This supports our initial conclusions as we can see that they also apply to situations where pre-critical and critical clusters are separated by more than one order of magnitude in size.
3. We note that for small cluster sizes there are ambiguities with the polymorph classification of interfacial molecules which is particularly relevant for the small clusters and leads to small deviations in the cubicity. We have visually verified that employing decomposition into double diamond and hexagonal cages (a much stronger topological feature) we see the same stark differences in the core-structure of critical clusters (panel c) and final simulation cells (panel d).

Overall, we think these new simulations greatly improve our manuscript by showing that our reasoning is valid for higher temperatures / larger size-difference between pre-critical and critical clusters and that therefore the differences in pre-critical fluctuations are significant according to what we discuss in the manuscript. We thank the referee again for this helpful suggestion. The developers of the enhanced sampling approach we used, Fabio Pietrucci and Silvio Pipolo, greatly helped us in performing these simulations and thus we think that they should be added as authors of the manuscript. The new figure is accompanied by a new paragraph:

To understand if our findings hold at higher temperatures we performed metadynamics simulations in our two systems at an elevated temperature of 235 K. This is around the highest temperature we can aim to study with our system size (as we expect the hom. critical cluster size to be ≈ 600 molecules [41]). In figure 3a we show the free energy profiles

obtained, where we note that the variable s describes the path from a liquid ($s \approx 1.1$) to a frozen ($s \approx 1.9$) simulation cell. We have employed an artificial soft wall at $s = 1.5$ to aid convergence for the region describing cluster sizes relevant to nucleation rather than growth. From these simulations we obtain (details in the SI) a free energy barrier on s_1 of 204 ± 5 kBT and on s_2 of 227 ± 5 kBT and critical cluster size on s_1 of 211 ± 11 and on s_2 of 104 ± 3 . Finding that $\Delta F_{s_1} < \Delta F_{s_2}$ and $n_{c,s_1} > n_{c,s_2}$ is entirely consistent with the trends obtained at lower temperatures. In addition it can be seen from figure 3b that the polytype of ice formed in s_1 and s_2 is not the same, the former being $\approx 55\%$ stacking-disordered and the latter being almost purely hexagonal. The deviations for smaller clusters are once again artifacts of the local order parameter employed at the cluster interface, where the classification is ambiguous. To illustrate the difference in the cluster cores, figure 3c shows representative snapshots for the critical clusters in s_1 and s_2 . We highlight in there hexagonal and double-diamond cages, the building blocks of I_h and I_c [43] that are a stronger topological feature than the local order parameter. In panel d) of figure 3 we illustrate that the substrate in s_2 avoids the stacking-disorder by stacking ice double-layers perpendicular to the surface which is a result of the crystal face (prism) in contact with the surface. We note in passing that this could be a general recipe for water and other tetrahedral liquids (e.g. group-IV elements or silica) and could also be exploited to design surfaces that nucleate pure cubic ice. Overall, the findings for the higher temperature agree with the simulations at lower temperature, suggesting that our reasoning also holds for situations where pre-critical and critical clusters are separated by more than one order of magnitude in size.

We also include details for these simulations in the methods section and a new section in the supporting information, laying out in detail how the metadynamics simulations were done, convergence tests and details about the committor analysis.

- In the interpretation of the results in connection with hetCNT, there are some weak points that need to be properly addressed. First, the authors completely ignore line tension effects, but this is not at all justified in the formation of such small clusters at large undercoolings. Small nucleation barriers can be dominated by line tension effects (see for instance Ref. 20). HetCNT taking into account line tension yields a functional form of the free energy landscape substantially different from Eqs. 1 and 2, and that may not retain the same functional shape. In addition, for small wetting angles, a pancake cluster would probably be more likely than a spherical cap (as mentioned by the authors in the main text and in the SI). The free energy expression for such a shape is different from Eqs. 1 and 2. Both effects (i.e. different line tensions and pancake-shaped crystals) may be dominant and could make unnecessary and meaningless the extension of CNT suggested by the authors.

This is another interesting point. The new simulations at higher temperature involve clusters that are much larger and not generally pancake shaped. The line tension effects are expected to decrease as we go to larger clusters, however the differences (barrier and crit. cluster size) that we observe at 235K have become more pronounced compared with the differences at 218K. We take this as strong indication that – whilst we cannot entirely rule out their presence – line-tension and shape effects are not the main driving force behind the stark differences that we observe, but it is the polymorph. Moreover we have clearly shown (new figure b,c,d) that the ice polymorph in the two systems is different and that in both systems the structure of the critical and pre-critical clusters are consistent.

We argue that the issue of the polymorph is a third, independent one that comes into play in addition to line tension and shape issues. One could argue that under conditions of weaker supercoolings and thus large clusters the issue of polymorph will be the most relevant of them as we expect e.g. line tension effects to cease for very large clusters while the difference in chemical potentials for metastable polymorphs could be very significant. Thus, in this work, we do not aim at assembling a comprehensive revised hetCNT that accounts for

all deviations that are known so far (line tension etc) and we rather focus on thoroughly pointing out the specific issue of different polymorphs.

We note that upon suggestion of referee#3 the section about the hetCNT extension has been shortened / made more phenomenological and the specific extension to hetCNT can now be found in the supporting information.

Regarding the main conclusions:

- The first paragraph is a bit confusing. The authors emphasize the fact that different pre-critical fluctuations lead to similar nucleation rates. But in the Methods, they clearly state "the substrate-water interaction... was tuned to achieve the same absolute nucleation rate". Since the substrate roughness, lattice spacing, and Lennard-Jones parameters of interaction with water are different, it is not surprising that interfacial energies, line tensions, and wetting angles (if they can be somehow defined for such small crystals) would be different and would lead to the preferential formation of small clusters with different structures.

We have tuned the nucleation rate to be nearly the same (at around 218K) to illustrate one of our main points: If the heterogeneous nucleation rate for two substrates is the same, hetCNT predicts the same enhancement factor for both. This is because hetCNT does not account for a possible change in polymorph compared to the homogeneous reference as there is no degree of freedom in the theory that allows for this. Regarding the different composition of small clusters we agree with the referee that the appearance of different pre-critical structures on the different substrates is unsurprising, but once again, then hetCNT would predict different nucleation rates. We believe the consequence of this has not been appreciated in the literature as we clearly show this different composition can translate into different critical and post-critical clusters. This leads to significant deviation from hetCNT as outlined in our manuscript. We have clarified this point in various places in the manuscript, e.g.

... Knowing the value of f_V for a given substrate is fundamental as it encodes all information about the nucleation enhancement which is reflected in the fact that all the curves in figure 1 retain the same functional shape and the steepness ratio

$\chi(f_V) = \Delta F(f_V) / n_c(f_V) = (f_V \cdot \Delta F_{hom}) / (f_V \cdot n_{c,hom}) = \Delta F_{hom} / n_{c,hom} = \chi$ (equation 3) is independent of the enhancement. ...

... In the framework of hetCNT this would imply that both substrates should have the same free energy profile and steepness ratio χ (equation 3), which is incompatible with the differences in pre-critical cluster sizes we observe. ...

- The first conclusion may be reinforced by the results of this work, but it is not new. Moreover, it is not fully demonstrated that the structure of the most abundant pre-critical polymorph is eventually the same as the one in the critical or larger clusters, for larger nucleation barriers and more realistic undercoolings.

We hope the referee is convinced by the new metadynamics simulations at higher temperature as they show the same trends compared to the original data and highlight that composition of small pre-critical clusters is the same as for larger critical and post-critical ones. Regarding the novelty of this point we believe it is important to mention that the selection of the polymorph can be facilitated during the early stages of nucleation. This means that substrates designed with the intent to facilitate a specific polymorph do not need to be iso-structural to this polymorph but they just need to have a low interfacial free energy with that polymorph - a much weaker requirement and potentially useful for structure-property screenings, or stacking-disordered materials such as ice in particular. Furthermore, we are not aware of studies that stress the meaning and potential use of pre-critical

fluctuations in the heterogeneous case for e.g. polytype screening or nucleation enhancement screening.

We changed the text of this first conclusion to:

Substrates can promote the formation of metastable phases by templating crystal faces that are unique to the respective polymorph. This is an extension of the rationale applied in experimental studies where iso-structural templates are used [45,46] since the substrate does not require the same structure but rather any structure that nucleates the right crystal face. In particular, for materials with different stackings (e.g. ice, group-IV elements or silicates), the templating of faces so that the stacking is perpendicular to the surface normal seems most promising and can avoid stacking-disorder.

- The second conclusion is not evident that holds for any undercooling and critical cluster size. There is the possibility that line tension and the non-spherical cap shape of the clusters will be more relevant than the differences between the bulk properties of the different polymorphs.

We believe our new data at higher temperature reinforce our reasoning and are a strong indication that - whilst we do not rule out their presence - line-tension and shape effects are not the main reasons for what we observe. Both effects are also expected to decline as we go to larger clusters (which we did with the new simulations), while the effect of different polymorphs will not cease even as we approach infinite cluster size. This also means that the difference in polymorphs is bound to play an extremely important role for more realistic conditions such as weak undercoolings, where we expect to deal with large clusters (and subsequently weak line tension / asphericity effects). We thus believe that the effect of polymorph selection in heterogeneous nucleation is important and should be added as a separate deviation from hetCNT to the ones that are already known (line tension, shape, etc.).

- The third one is not well substantiated. The most probable largest pre-critical crystal cluster is strongly affected by finite size effects, and depends crucially on the total number of molecules N used in the simulation and the temperature (the larger the N and the smaller the temperature, the larger will be the size of the most probable largest cluster). In addition, the different probabilities of having different polymorphs for small fluctuations might be irrelevant in the formation of the critical cluster, for low (and more realistic) undercoolings and correspondingly large critical clusters.

Our new simulations show that the pre-critical fluctuations correspond to the critical and post-critical clusters even at higher temperature, which was one of the main concerns of the reviewer. This means that pre-critical fluctuations do indeed carry information about the nucleation event as demonstrated in our work. We think that this information will be useful in qualitative studies and perhaps even quantitatively (formula 2). However, the reviewer rightfully mentions a number of complications that need to be sorted before this can be done.

We think the fact that the largest cluster is system-size dependent should not be termed a finite-size effect, which is a term usually used to describe the presence of unphysical self-interaction due to the size of a simulation cell. This unphysical self-interaction is actually much better avoided for pre-critical clusters than for critical ones as they are much smaller. Rather, the size dependence of the biggest cluster should be seen as an extensive property such as volume etc. This makes it easy to choose the settings that make an examination of pre-critical clusters comparable, i.e. choose the same contact area in het. simulations at the same temperature etc. Furthermore, one could also study the statistics of all clusters in the

system rather than only the biggest one, which then would be a system-size-independent statistics.

In any case, a quantitative assessment of pre-critical fluctuations needs care (and possibly also more data than the two substrates in our study) and thus is beyond the scope of our work. We still want to mention the potential capabilities (we “hypothesize”) in our third conclusion point since this might facilitate studies of pre-critical fluctuations across the field.

We added to this conclusion:

... provided one pays attention to the comparability of systems (same contact area, temperature etc.).

Finally, some minor issues:

- Eq. 6 is missing a minus sign, and the need of the parameter γ is unclear; in a true nucleation (activated) process, the survival probability is exponential.

The typo was fixed accordingly. The γ extends the applicability of the formula to processes that are more of a relaxation rather than an activated process. The use of this as opposed to fixing $\gamma=1$ can be seen as a test that probes if we are looking at activated processes or if the supercooling is too strong (spinodal events). The resulting survival probabilities from our simulations are perfectly exponential-like (see SI Fig. S1) and all fits yield a γ so that $1 < \gamma < 1.1$, which means we are assured that we are looking at nucleation rather than relaxation. We now note in the SI:

Indeed, all four fits yield values of γ so that $1 < \gamma < 1.1$, which means that even at this strong supercooling we are looking at activated processes rather than relaxation.

- Since the mean first passage time seems to be evaluated in the reconstruction of the free energy, why was not used to get an estimate of the values of nucleation rates?

In our experience, the calculation of the rate via the mean-first passage time needs much more statistics (and disk space for the cluster analysis in each frame of each trajectory) and has higher uncertainty than the approach via the survival probability, which is why we used the latter.

- It would be helpful to report also the structure or composition of the critical clusters, as it has been done for the pre-critical clusters in Fig. 2a.

We did not report numbers about the composition of critical clusters because we have much less statistics for them compared to the pre-critical ones. However, simply put, the ones we can probe have nearly identical composition as the pre-critical clusters of the respective surface, i.e. s_1 has stacking-disordered pre-critical and critical clusters and s_2 has nearly exclusively hexagonal pre-critical and critical clusters. This is also also part of the reason why we believe that pre-critical and critical clusters are deeply connected as outlined in our manuscript and also supported by the new simulations provided. We note this now in the text:

Although the statistics we have for critical clusters are worse, we note that on each surface their composition was nearly identical to the one of the respective pre-critical clusters. This further suggests that there is a causal connection between critical and pre-critical clusters.

- The asphericity of clusters may strongly depend on their size. It would be also interesting to have a plot, similar to Fig. S3, but in terms of cluster size rather than distance from the substrate.

Indeed the asphericity will be size dependent as well. However, we do not encompass a large size range of clusters and therefore do not expect major differences or indicative trends that could explain our observations.

In summary, although the ideas and results presented in this work are potentially interesting, there are many loose ends that the authors need to address doing a major revision before the paper can be recommended for publication.

We hope our revisions address the points raised by the referee and thank them once again for their comments as we believe their suggestions have allowed us to greatly improve the manuscript.

Reviewer #3 (Remarks to the Author):

This manuscript reports what is a rather interesting study of factors important to heterogeneous nucleation. Employing molecular dynamics simulations, the authors investigate the influence of two model surfaces on the nucleation of ice, where the surfaces have been parameterized to give essentially the same enhancement of the nucleation rate (relative to bulk). The authors then demonstrate that the pre-critical (structural) fluctuations are notably different for each system and that heterogeneous classical nucleation theory (hetCNT) is unable to account for this behavior. This work thus provides important insights into the possible connections between specific structures (appearing prior to the nucleation of the system) and the thermodynamics and kinetics of crystal nucleation; as such it is potentially of great interest and broad implications.

The results of this manuscript lead the authors to identify two primary conclusions (where each is discussed in its own section). The first of these is that the formation of different polymorphs leads to different heterogeneous free energy profiles. I generally found this section well supported and an interesting read. However, I found the second section, which describes an extension to CNT to account for the observed differences, much less convincing. Consequently, I would recommend that the authors either scale-back considerably their second section (on the Extension of CNT), or that they significantly improve their justification of this approach and more clearly describe its assumptions/limitations (further details are provided below). Hence, I would support the eventual publication of this work once the authors have addressed this major concern, along with the other issues I have detailed below.

We thank the referee for carefully reading and assessing our work. We have followed the referee's suggestion to scale back the mentioned section and shifted the specific discussion of the extension in the framework of hetCNT into the SI. Furthermore we provide new simulation results at higher temperature which further strengthen our initial conclusions. Detailed changes are outlined below:

1) I found the section discussing the Extension of CNT to be the weakest part of this manuscript - hence my recommendation is that it either be appropriately justified, or be made a simpler, more phenomenology discussion (I would tend to favor the latter).

a) New relationships for describing heterogeneous nucleation are introduced on page 7. However, the authors do a poor job of outlining clearly the assumptions on which these expressions are based (although afterwards, statements such as "provided the approximations of CNT as reasonable" appear).

We have scaled back and tried to better explain our extension of hetCNT, As it happens our approach does not introduce any further assumptions compared with hetCNT and thus comes with all of its weaknesses and strengths. We added the following to the paragraph about the hetCNT extension (now found in the SI) to clarify the scope and limitations of our approach as suggested by the referee:

We note that in this work it is not our aim to include corrections for several of the already known possible shortcomings (e.g. neglect of the line tension) but rather we focus solely on how to account for a change in polymorph induced by the substrate.

...

The CNT assumptions implied are: i) the nucleus has spherical cap shape, ii) thermodynamic properties of small clusters are assumed to be the values of the bulk and iii) a well-defined surface that separates cluster from liquid.

...

Note that our extension has not introduced any further assumptions, but we have solely used the tools supplied by CNT to illustrate how a change in polymorph needs to be included in the theory.

b) These relationships assume that properties of the bulk phases (i.e. chemical potentials and surface free energies) remain valid when applied to crystal nuclei. In the present case, the two systems being studied exhibit critical nuclei consisting of about 20-30 water molecules. I might hope that the authors suspect that there are major limitations in assuming these macroscopic properties are applicable to such small (nanoscale) clusters. Indeed, given that the critical nucleus for the corresponding homogeneous process is considerably larger (hence more likely to be reasonably described by macroscopic parameters), perhaps the breakdown of these assumptions (which become unique to each system) is also a major part of the observed phenomenology?

Indeed we agree with the referee that the size dependence of thermodynamic quantities is a major problem when theoretically describing brute-force events with small clusters. To address this issue we have performed new simulations at higher temperature (235K) achieved with metadynamics utilizing permutation invariant vectors (PIV) as order parameter. All details can be found in a new chapter in the SI and a new section in methods. Our results are shown in a new figure accompanied by a new paragraph describing the results (see response to referee#2 on page 2-3 of this reply).

These simulations are costly, but the critical clusters at 235K are at least one order of magnitude larger and their properties will be more bulk-like. Thus, if this would have caused our observation we would expect there to be less or no differences between the systems at higher temperature. However, we find even more pronounced differences regarding the barrier and critical cluster size between the two systems. We take this as indication that - while there might be an effect of cluster-size dependent thermodynamic properties - this is not the main force behind the stark differences that occur.

Furthermore, we have followed the referee's suggestion to scale back on the discussion about the hetCNT extension. We shifted the section deriving the formulas into the SI and add to the discussion in the main text two paragraphs that emphasize the main conclusions we draw. Instead of arguing in the specific framework of hetCNT we make clear the main correction that must follow from polymorphism in heterogeneous nucleation for any theory: the correct homogeneous reference must be chosen as otherwise the enhancement factor becomes ill-defined. The added/changed text is:

We now try to place the results of this study in a broader context and discuss some of the implications of our findings. The first consequence drawn from the possible occurrence of different polymorphs is that the fundamental result of hetCNT that reads $n_{c,h\text{et}} = f_V \cdot n_{c,h\text{om}}$ and $\Delta F_{h\text{et}} = f_V \cdot \Delta F_{h\text{om}}$ is not true for cases where the substrate promotes the formation of a polymorph different than the one that is formed homogeneously. This is because the enhancement factor f_V is only properly defined if the het. quantity it describes refers to the hom. reference of that polymorph. In general when thinking about het. Nucleation there are three possible ways to account for the enhancement factor: (i) An expression in terms of a shape factor $f_V = V_{h\text{et}} / V_{h\text{om}}$; (ii) An expression in terms of a nucleus factor $f_N = n_{c,h\text{et}} / n_{c,h\text{om}}$; and (iii) an expression in terms of a potency factor $f_P = \Delta F_{h\text{et}} / \Delta F_{h\text{om}}$. These three definitions are equivalent under the assumption that they describe events where the same polymorph has been formed. However, if different polymorphs are compared the concept of the enhancement factor becomes ill-defined. We derive in the SI correction factors in the framework of hetCNT that account for this change. The fact that increasing the temperature accentuated the difference in the free energy profiles observed on the two substrates is an indication that effects like line tension [44] and

cluster asphericity [12] are not the main reason for our observation (as those likely decrease with increasing temperature / increasing cluster sizes), but rather it is caused by the different polymorphs. Hence, we believe that the polymorph is a separate issue that should be taken into account in a comprehensive (het.) nucleation theory, in addition to known shortcomings of CNT or its het. extension. We speculate that for the same reason the polymorph could even be the most relevant deviation from hetCNT at high temperatures.

Another implication of this work is that pre-critical fluctuations are comparable for different substrates only if compared to the correct hom. fluctuations of their corresponding polymorph. In our study, the comparison of the pre-critical fluctuations of s1 and s2 with the hom. case would have resulted in the conclusion that s1 enhances the nucleation and s2 does not (from figure 2a), while they actually lead to nearly identical enhancement. We have illustrated this in figure 4a-b where we draw hom. and het. nucleation profiles for two different polymorphs. The grey shaded area and how far it stretches on the x-axis illustrates what cluster sizes can be reached through thermal fluctuations. This ultimately determines the extent of pre-critical fluctuations and is very different for the two polymorphs as a result of their different hom. free energy profiles. Upon comparing to the hom. nucleation of a single (homogeneously dominant) polymorph (which without loss of generality we assume to be hom,1 in figure 4c) the apparent discrepancy becomes clear.

c) In view of the above, it is then crucial for the authors to demonstrate that their new relationships (on page 7) can be used to describe (fit) the behavior of the current systems.

We find that our correction is able to qualitatively reproduce a main finding: Since the correction terms for barrier and critical cluster do not scale by the same factors, the resulting ratio between barrier and critical cluster can change for different het. nucleation events if there are different polymorphs. hetCNT instead predicts this ratio to be constant and even independent of the enhancement and therefore the ratio should be the same for s1 and s2. The fact that we find otherwise from our simulations can be explained by our correction on the basis of different polymorphs.

There are two main difficulties with directly and quantitatively assessing the introduced extension. 1) CNT and hetCNT describes the nucleation as a function of an isolated cluster n , rather than the biggest cluster N_{cl} in a system of given size. We have tried to convert the free energy profiles from N_{cl} to n , however find that to the best of our knowledge there is no rigorous way of aligning the profiles at $n=0$ which has a crucial impact on the obtained barriers and thus the comparison. 2) we are not aware of reliable estimates of the chemical potential difference for I_{sd} at the given temperature and also for the specific cluster size of our study. Thus, we followed the referee's suggestion by shifting the specific argument about the hetCNT extension to the SI but keeping the main conclusion that the correct hom. reference has to be chosen in any theoretical description in the main text.

i) It is not obvious how these relationships were used to arrive at the sketch provided in Figure 3 – specifically what assumptions were made for the appropriate parameter values?

In Figure 3 we illustrate the connection between het. and hom. free energy profiles for different polymorphs (a,b) and that they belong to the same functional family within each polymorph. But most importantly c) shows how a comparison between hom. and het. events that do not nucleate the same polymorph can cause the apparent difference in pre-critical fluctuations, even if the nucleation rate is the same. Also, in there we can see that the scaling compared to the hom. case of the het,2 profile is not by a single factor as het,1 but the critical nucleus size and the barrier are scaled differently. Overall, this sketch qualitatively resembles our simulation results and is intended to illustrate the reasoning in the text, but we do not plot any fit or parametrized model. To clarify this, we changed the caption to:

Schematic illustration of the connection between free energy profiles, pre-critical fluctuations and polymorph: a-b) Resulting het. free energy profiles for two different polymorphs which belong to the same functional family as their hom. reference. The extent of thermal fluctuations is indicated by the grey shaded area. c) Observation in a simulation or experiment where the hom. nucleation of the dominant polymorph 1 is compared to het. nucleation events that form the same (het,1) and a different (het,2) polymorph. While for the het,1 profile the crit. nucleus and the barrier are scaled by the same factor (as predicted by hetCNT), these are scaled differently for the het,2 profile when compared to hom,1. Note that the graphs are qualitatively equivalent to our simulation results, but do not result from a fit or parametrized model.

ii) Reasonable estimates for the difference in the free energies of the two (bulk) ice phase are known (see for example theoretical predictions by Molerino and co-workers, or estimates from experiment by Jahari). Additionally, since the ice phases differ only in their stacking (of hexagonal layers), we would expect their (bulk) densities to be essentially identical. It should then be possible to use these values to test the validity of their expressions.

As noted above in the responses to point 1c) there are several other issues with quantitatively assessing the relation. Thus we think a quantitative assessment is reasonable, and our corrections do indeed have the capability to reproduce the difference in shape (or equivalently the ratio between barrier and critical cluster size as defined in the new manuscript).

2) Some additional minor issues:

a) On page 3, second to last sentence of the first paragraph: I found the wording here confusing.

We meant to say that the formulas and concepts of hetCNT are useful to illustrate our findings and conclusions on pre-critical fluctuations. However, our results and interpretations fit easily in any other theory (that is not necessarily CNT based) that also describes the free energy profile.

b) On page 3, last sentence of the first paragraph: claiming “have never been studied” is perhaps a bit of an overstatement here. Others have certainly looked at structural fluctuations prior to nucleation (e.g., for ice, consider the work of Debenedetti – ref. 47, and for gas hydrates the work of the groups of Kusalik and of Molinero). What is important in the current work is the characterization and the degree of distinction. Appropriate clarification should be added.

We intend to emphasize that the role of pre-critical fluctuations in the case of heterogeneous nucleation (as opposed to several studies for the homogeneous case) is less well understood. We changes the corresponding sentence in the text to:

Although many aspects of nucleation have been studied in great detail, the role of pre-critical fluctuations in het. nucleation is less well understood. However, deeper understanding could potentially be exploited to gain insight into fundamental aspects of het. crystal nucleation.

c) Page 5: the authors use the terms “identical” (line 2) and “the same” (second line from bottom), when I think they mean “essentially identical” and “essentially the same”.

Changed accordingly in the text.

c) On page 5, line 8: the term “of each frame” should be clarified.

Changed to:

In figure 2a we plot the size distribution of the biggest ice-like cluster that can be found in each snapshot from the trajectories.

d) On page 6, line 7: the authors state “suggests an additional degree of freedom” when it would be more correct to state “suggests that at least one additional degree of freedom”. (There are likely deeper issues associated with choices of order parameters here, but such a line of inquiry would be beyond the scope of this work.)

Changed accordingly in the text.

e) On page 6, second paragraph: in this paragraph, the authors indicate that formation of Ic layers is “strongly disfavored” for the s2 surface, yet at the end of the paragraph they report that 90% Ih clusters (which implies 10 % Ic clusters). Apparently the level of selectivity of the structural fluctuations appears not to be extremely selective (rather there seems to be enrichment of Ih clusters). Either the authors should adjust their language, or explain the discrepancy.

We understand this as a side-effect of the classification of single-molecules into Ih and Ic like molecules. At the interface between core molecules and liquid molecules the Ih or Ic character of them is not very well defined and it can happen that we classify a molecule on the outside of an otherwise purely hexagonal cluster as cubic. One could argue about classifying them as interfacial rather than Ic or Ih like. Considering the small size of clusters in the brute-force approach this can make for a percentage of <10% being Ic like (s2) but never reaches the 60% that we get for clusters that are in their core stacking-disordered (s1). Indeed, we have never observed a single properly stacking-disordered cluster at the substrate interface for the s2 system which stems from the specific crystal face that forms there being incompatible with stacking disorder as mentioned in the text. We added a clarification the the text:

The apparent ~10% Ic -like molecules in s2 is are to uncertainties in classifying interfacial molecules at the edge of the cluster. We have visually verified that, in contrast to s1, in s2 we never observe clusters near the substrate that are in their core stacking disordered (see also figure 3c).

f) On page 7, second line after Eq. (5): “This explains why ...” is not appropriate here (since other explanations are possible). More appropriate to state “This may account for why ...”

Changed accordingly in the text.

g) Sentence beginning at the bottom of page 7: I found this sentence rather confusing (in part because of the use of the word “definitions”). I would suggest rewording (e.g. “In general when thinking about het. nucleation there are three possible ways to account for the enhancement factor: (i) through an expression in terms ...”)

Changed accordingly in the text.

h) Reference 48 should be removed, or a publication provided.

Changed accordingly.

We thank the referee once again for their constructive suggestions and believe that in addressing them our manuscript has improved.

Reviewer #2 (Remarks to the Author):

The authors have made an important effort to dissipate any possible doubts and to clarify the potential weaknesses of their work. In particular, their new simulations at high temperatures confirm and reinforce their conclusions. The response letter is also clarifying and well substantiated. Accordingly, I am glad to recommend the publication of the manuscript in its present form.

Reviewer #3 (Remarks to the Author):

The authors have addressed all my previous concerns. I recommend publication.

I did note the following typo in their revised text on page 8:

"The apparent ~10% Ic -like molecules in s2 is are to uncertainties ... "

Reviewer #2 (Remarks to the Author):

The authors have made an important effort to dissipate any possible doubts and to clarify the potential weaknesses of their work. In particular, their new simulations at high temperatures confirm and reinforce their conclusions. The response letter is also clarifying and well substantiated. Accordingly, I am glad to recommend the publication of the manuscript in its present form.

We are delighted that the referee recommends publication for our revised manuscript and thank him once more for his constructive suggestions.

Reviewer #3 (Remarks to the Author):

The authors have addressed all my previous concerns. I recommend publication.

I did note the following typo in their revised text on page 8:

"The apparent ~10% Ic-like molecules in s2 is are to uncertainties ... "

The typo was fixed accordingly. We thank the reviewer for their recommendation and their helpful assessment of our manuscript.